# Exploring the Impact of Complex Multi-Level Governance Structures on the Societal Contribution of Universities to Knowledge-Based Urban Development

**Savis Gohari** *[ID], **Tor Medalen** and **Rolee Aranya**

Department of Architecture and Planning, Norwegian University of Science and Technology (NTNU),
7491 Trondheim, Norway; tor.medalen@ntnu.no (T.M.); rolee.aranya@ntnu.no (R.A.)
* Correspondence: savis.gohari@ntnu.no; Tel.: +47-94484660

**Abstract:** The current debate aims to reconceptualize the changing role and missions of the university in today's knowledge economy and investigate how universities' knowledge resources can benefit urban development and inform the direction of changes in universities. However, there is a lack of empirical studies exploring how governance networks and the institutional conditions of universities in specific contexts can support, limit and/or incentivize the integration of academic activities into societal development. There is a discussion of the various and paradoxical components of university transformation (institutional and physical), affecting their societal contribution, which conceptualizes a holistic and integrated approach towards governance that previously has not been fully investigated. This paper will examine the co-location case of university campuses in Trondheim to explore the implications of a multilevel governance network for achieving the goals of sustainable and knowledge-based urban development. This paper suggests that engineering effective governance is challenging and that factors related to the culture of the institution and their connecting strategies, government priorities, and temporal factors have a great influence on universities' contribution to their societies. While investigating governance in this topic requires political, cultural, and periodic review, focusing on the interactions of governance multi-layers, this paper concludes that governments' control functions or some moderate hierarchical coordination is necessary to avoid the failure of university governance and unbalanced societal contributions.

**Keywords:** university; knowledge-based urban development; governance; structure; societal contribution; co-location; Trondheim

## 1. Introduction

*Gaps in the Knowledge and the Research Question*

In today's rapidly globalizing world, 'knowledge', along with the social, technological, and economic settings, is seen as today's economy (Yigitcanlar et al. 2007; Gabe et al. 2012). The process of transforming cities into knowledge cities is influenced by universities that help cities to embed their knowledge into sustainable development. However, such a process of transformation has been an intricate evolution that has brought different social, economic, and ecological interests into conflict with political interests at different societal levels. On the one hand, universities have been increasingly called upon to use their knowledge assets to contribute to local, urban, and regional development goals (such as sustainable development, smart cities, and knowledge-based urban development), being active partners in urban governance arrangements. On the other hand, universities' activities and

policies are regulated and managed in more extensive networks of national regulation and market steering (Benneworth et al. 2017). These European national regulations are also increasingly influenced by the European policy framework, which stimulates universities to prioritize research excellence and internationalization (Karseth 2008). The latent conflict between these diverse interests and strategies at different political and power levels can lead to prolongation, recurring controversies, stagnation, and moments of adaptation in the development processes of universities.

Different studies have highlighted the tensions existing at different layers of governance. For example, Dill and Helm (1988); Kezar and Eckel (2004); Trakman (2008); Clark (1986); Amaral and Magalhaes (2002) have focused on internal university governance, i.e., the understructure level, while Benneworth et al. (2010); Perry and Wiewel (2005); Russo et al. (2007); and Vázquez et al. (2008) have focused on urban governance, i.e., the middle structure level, and Christensen (2011) has focused on the national-international governance network, i.e., the superstructure level. Many studies also indicated a relationship between these levels, though there have been few attempts to investigate the dynamics of interactions and mechanisms between these levels. Thus, their relationships have not been explicitly explored. For instance, Benneworth et al. (2017) discussed how national governance policies can influence and challenge the universities' urban governance and thereby, their contribution to regional development. However, they did not explore how universities' internal governance networks can contribute to the resolution or indeed the exacerbation of these influences and challenges.

On the other hand, while the impact of knowledge assets at different levels, particularly in regional development, receives much attention, few studies have focused on governance in planning and decision-making processes regarding the interrelationships between academics, city planners, and governmental or political authorities/actors at different levels in such processes. Much research has explicitly focused on the real costs and conflicts of interest involved in exchanges between university and other related actors. For example, Benneworth et al. (2010) attracted attention to the fact that cities and universities may fail to achieve mutual benefit due to a misalignment and disagreement of their strategies. They may have contradictions regarding university expansion plans, local transport, and housing infrastructure that can impose barriers on either side. Russo et al. (2007) used a consensual and deliberative approach to emphasize collective, pragmatic, and participatory problem-solving in university-development processes. They proposed that joint management should not include merely the university and the city but also students, academic community, entrepreneurs, and citizens, whose personal ties and informal networking have effects on university development procedures. However, Benneworth et al. (2010) and Russo et al. (2007) both largely neglected the role of governmental and political authorities as well as deeper organizations' cultural involvements in the change process that underlie the interdependency between different levels of society. More importantly, they failed to take account of the strategic role of main actors and the interactions of their interests and power relations. Etzkowitz and Leydesdorff (1997) considered the government's role and introduced the triple helix model, emphasizing the interaction between university, industry and government as being the key to societal innovation and resultant economic growth. However, they did not consider the various ways these three categories of actors can link and interact with each other at different levels to support societal contribution. In many studies, the focus has been only on one or several limited levels of governance. Therefore, the main research question was '*how do the multi-level governance networks influence the university development processes and its societal contribution?*'.

The analysis of existing studies in the governance topic usually takes place at a high level of aggregation and generalization, which obscures the diversity of individual cases. Accordingly, there is a need to look at the governance of university development in its multi-level (i.e., understructure, middle-structure, and superstructure) and complex context to understand which actors, institutions, processes, and relational mechanisms at different levels have had an impact on the implementation of university development. In this regard, this paper adopted a single case study in Trondheim to investigate governance in its complex, uncertain and multi-level system.

In order to answer the research question, the rest of the paper is structured as follows: Section 2 deals with the literature review and the contextual background of the new knowledge-based urban development (KBUD) within both (inter)national and local contexts and highlights the necessity of network governance for redressing the balance between such changes and the original identity of universities. Section 3 describes the methodology of this research, which is a qualitative case study, and provides a detailed explanation of the methods used in this paper. Section 4 describes the case of co-location within the governance framework over time. Section 5 analyzes the interactions of network governance at three levels: understructure, middle structure and superstructure, understanding how conceptual components explain (or do not explain) the specific outcomes of the case, thereby answers the research question. Section 6 synthesizes the findings, the analysis, and the implications of the findings for future studies.

## 2. Literature Review

### 2.1. From 'State' and 'Market' to 'Governance'

Many theorists argue that the emergence of 'governance' reflects the change in governments' roles and styles of governing (Klijn 2008). As Berger (2003) argued, "the debate about governance arose when the traditional forms of government intervention and policy making was questioned".

Sørensen and Torfing (2004) explained how the modes of governing gradually shifted away from the traditional hierarchical control of the government, in which the political values and preferences of the government were translated into laws and regulations and were implemented and enforced by publicly employed bureaucrats. As Sørensen and Torfing (2004) mentioned, in the 1970s, public choice theorists blamed this mode of bureaucratic welfare state for not being democratic, inexpensive, or efficient. Instead, a 'less state and more market' mode of governing was introduced, in which the invisible hand of the market could not only ensure an optimal allocation of private goods but could also regulate the production of public goods more efficiently. Therefore, according to Rhodes (1997) and Pierre (2000), the new growing interest was in forms of economic coordination that conformed to neither pure markets nor to unitary hierarchies/government. Rhodes (1996) explained that the shift in emphasis from hierarchies to a more competitive basis for providing public services, manifesting the theory of 'new public management' (NPM), implied two meanings: managerialism and new institutional economics. The former indicated bringing private and public sectors together, and the latter introduced incentive structures, such as market mechanisms, into public service provision. However, as Jessop (1998) argued, the increased reliance on market forces was also criticized for depoliticizing public governance and for failing to prevent instability, externalities, and inequality, and thus, for enhancing state control (rather than reducing it) (Sørensen and Torfing 2004; Jessop 1998; Sager 2013).

According to Sørensen and Torfing (2004), both 'state' and 'market' modes of governing failed because they threw their weight behind power and money, and thus undermined the social bonds and virtues of civil society. Therefore, a new governing model was demanded to encompass the three dimensions of (i) the state, (ii) the market, and (iii) civil society. This argument is consistent with Scharpf (1994), indicating that the concept of 'governance' emerged to suggest that decisions should no longer be enforced by legal measures, economic incentives, or normative control alone. Accordingly, as Murdoch and Abram (1998, p. 41) indicated, governance suggests "a shift from state's sponsorship of economic and social programs to partnership arrangements between both governmental and non-governmental organizations".

Many arguments focus on the minimization of governments' traditional role as a result of economic globalization. This paper agrees that although the form and level of government's power and influence has changed, it has not been taken from government and given to others. Indeed, it is partly shared with other actors in policy networks. Therefore, instead of seeing governance as a "hollowing out of the state" (Rhodes 1996), we consider it as a new strategy for state re-structuring

based on a public-private coordination (Pierre 2000) with the plurality and complexity of hierarchies, markets, and networks (Kjaer 2004). Subsequently, the meaning of governance network (in this paper) is not exclusive to informal and horizontal relationships but also includes hierarchical authority patterns. The chosen definition of 'governance' is "the ways in which stakeholders interact with each other in order to influence the outcomes of public policies" (Bovaird 2005, p. 220) and the system through which a kind of order is achieved among several actors who are cooperating and contributing resources to the negotiation process (Jessop 1998) to reach a compromise, even though they might have conflicting interests.

According to Newman and Thornley (2002), governance capacity and planning systems in cities are different and restricted by national institutional differences. Even in Europe, they saw significant differences between east and west and between north and south. For instance, they mentioned that in Britain, the Thatcherite ideology of economic liberalism and the 'authoritarian decentralist' approach have resulted in an enhanced role of the private sector and have weakened the importance of local democracy (Duncan and Goodwin 1988). In Norway, many scholars agree that there currently is a fragmented management structure in which a multitude of actors are involved in urban management, while there is no strict judicial hierarchical binding between the governmental levels. As Harvold and Nordahl (2012) argued, this introduces some freedom and flexibility into the Norwegian system, although the hierarchical/government's logic still plays a dominating role in regional-local development (Hanssen et al. 2011). Therefore, this paper acknowledges that in order to understand governance, the governmental priorities and power structures within a multi-level system should be investigated.

### 2.2. Similar Transformation Processes within the Higher Education Governing System

The same process of governing changes is emerging in the higher education system in many Europian countries. Amaral and Magalhaes (2002, p. 2) explained that at the end of the 18th century and the beginning of the 19th century, the university was "an agent of national reconstruction, allied with the overhaul of recruitment to the apparatus of state". In addition to developing skills and human capital, universities had to forge a national political identity through the preservation and enhancement of the national culture, thereby consolidating the nation-state (Amaral and Magalhaes 2002). The university was placed within the public domain as a national responsibility, which limited the university's control and administration during that period. "The state acted as the sole regulator of the higher education system by using traditional mechanisms of public regulation, including legislation, funding, and in many cases, the appointment of professors" (Amaral and Magalhaes 2002, p. 3).

Later, according to Collini (2012), governmental intervention and regulation were perceived as being excessive, and the governments' centralized and detailed control over university systems was criticized. In response to the criticisms, government strategies changed by partially replacing the traditional public regulation mechanisms with market-type mechanisms and progressively embracing the principles of university autonomy and self-regulation. Neave and Van Vught (1994) saw this transition as a shift from the model of state control to the model of state supervision. These new mechanisms in Europe induced competition among universities (for students, for funding, for research projects, etc.) to become more efficient and more responsive to outside demands. This process led to the emergence of a new university organizational stereotype or model known as the 'entrepreneurial university' (Marginson and Considine 2000). These new changes and outside pressures on universities were assigned to the nature of the university as a public good, which required its interaction with and response to outside interests for its development. It is in this context that the concept of the 'third party' or 'external stakeholder' emerged in many university management structures to make universities more visible to society. The presence of the third party was considered both legitimate (having a 'legitimate' interest in the social, economic and cultural function of the university) and useful (enhancing the university's innovation and responsiveness to the 'real' needs of society) (Amaral and Magalhaes 2002, p. 2).

Later, universities were appointed to be even more responsive to the needs of society, which required the involvement of society (external representative) more directly in the university's internal affairs, such as the proposed budget, localization, and priorities of the study programs. Thus, the third party's responsibility to protect academic freedom from external interests has gradually changed to protect outside interests from attacks coming from inside the institutions themselves (e.g., opening up university campuses and facilities to the outside world). By eliminating the collective decision-making tradition and replacing it with a more managerial organizational structure, many universities' superstructure governing model and relationships with the government are transforming. This includes their "systems of decision-making and resource allocation, the patterns of authority and hierarchy, and the relationship of universities to the different academic worlds within and the worlds of government, business and community without" (Marginson and Considine 2000, p. 7). Exploring such changing relationships in countries like Norway, in which universities, as part of higher education, are state-run, is more critical than in private universities.

### 2.3. The Urban Role of Universities in the New Knowledge Economy

In light of universities' educational mission, physical location, economic relations, and political demands, they have an important and complex role in the development of their host city. In the past, universities defined themselves by their elitism and their isolation. However, today's economies require that knowledge be distributed and widely accessible. Therefore, universities can no longer be isolated. Indeed, "in a healthy knowledge society, the university becomes the city, and the city becomes the university" (Corneil and Parsons 2007, p. 115).

One aspect of universities' societal contribution to urban regeneration appears to depend on the integration of their campus with the urban fabric and city life to increase and encourage cross-encounter and informal communication rather than simply offering education and research. For instance, Columbia University in Manhattan has started to give lectures in shop premises at the street level so that everyone can attend classes (Hajer and Reijndorp 2001). The Honggerberg campus of the Swiss Federal Institute of Technology in Zurich (ETH) in Zurich, which was originally built as an isolated/mono-functional island, has become interwoven with the city through a densification of the campus with commercial functions like shops, cafes, and restaurants. Lecture halls and foyers are also being used for events and community activities, which implies a sociocultural exchange with the city (Christiaanse 2007).

In addition to integrating university campuses with city activities, the geographical proximity of campuses with business or other knowledge institutions, forming a knowledge precinct/hub or 'science park', is another way for universities to contribute to societal and economic development. The co-location of university departments or campuses is also consistent with increasing the geographical proximity to enable easier and more frequent interaction between members of different departments. The underlying assumption is that by bringing individuals together, the functional barriers that separate fields of knowledge are broken down, thus promoting close interaction to achieve common goals/interests.

Nevertheless, the power relations underpinning the urban-regional development and the university development can be very different and challenging (Benneworth et al. 2010). While the city authorities may see their interventions in the university development processes necessary for improving the spatial quality or the competitiveness of their locality, universities may expect the local authority to only play a reactive and subordinated role (Benneworth et al. 2010). There are also other groups and individuals who have specific (self-)interests in the university development process, and their actions may lead to deep shifts and changes in the way university strategies and decisions are developed and employed. Thus, the universities' development can be key political and social elements of "urban partition and conflict" (Perry and Wiewel 2005). By considering university development as a political procedure, there is a need to understand the local politics and rules of the game in

university and urban governance processes (the middle-structure level); otherwise, the possibilities of the development carrying forward will be challenging.

### 2.4. University Internal Governance in Response to the External Environment and Demands

In many of the Continental European states, the government/ministry of education has the regulatory and funding responsibilities for higher education. The relationship between society and higher education in the USA, Canada, Australia, and the UK is driven by a form of academic capitalism that relies on market-type interactions (Slaughter and Leslie 1997). However, in Europe, market elements are less radical and far-reaching and 'network' types of relationships in which the state plays an important role are more present (Maassen 2002). In this matter, the allocation of public funds is the main factor that determines the university governance and the academic and administrative decisions.

University governance (at both superstructure and understructure) reflects the institutional and ideological contexts of universities that influence their organizational development and the work experiences, norms, and practices of academic scientists (Trowler 2001; Lam 2010). Until a few decades ago, many universities were seen as a closed community or 'ivory towers' in which the academic community needed to be kept away from unwanted distractions (Amaral and Magalhaes 2002; Barrett 1998). Higher education was only reserved for the minority of the population. The spatial and physical quality of universities also reflected the sense of their being a part of the elite. Many university campuses, such as Yale in the USA and Cambridge and Oxford in the UK, were supposed to separate the academics' lives from the outside world, depicting a "relatively closed group with particular intellectual, social, and political qualities; a privileged, dominant social stratum" (Deplazes 2007, p. 41). On the other hand, with a reference to the dominant role of students in the French revolution in 1968 and Iranian revolution in 1979, and at UC and Berkeley (1960s) in instigating political unrest, students were seen as a 'political force and a potential danger' and keeping them in a closed community was done for governments' own safety (Deplazes 2007). Accordingly, universities traditionally used to steer themselves in academic matters. However, as mentioned in Section 2.2, in the nineteenth century and under the influence of the industrial revolution and the emergence of social awareness, the university model as an isolated community was thoroughly revised. In addition, the underlying assumption of the current international reform ideologies is that academia is poorly equipped for running universities alone and contributes to their institutional success and competitive power (Maassen 2002).

Nevertheless, the recent changes taking place within higher education have faced a certain amount of skepticism, with some suggesting that the knowledge-economy revolution has threatened the academic freedom and autonomy/self-steering model and traditional norms and values (Lam 2010), and has replaced them with short-term economic views and criteria (Amaral and Magalhaes 2002). In addition, some of the existing internal tensions and conflicts of interests are seen to be a result of external environments/demands. Accordingly, academics have tended to leverage these tensions and conflicts of interest for their academic freedom, reflecting path-dependent resistance towards any change or reform imposed on them or the university (Huisman 2009).

The question of how universities (and their internal governance structures) should react to these external pressures to increase their responsiveness to society has still not been discussed. All the arguments above bring attention to the topic of governance in understanding how universities should react to balance the internal and external demands for change and stability. Oliver (1991) argued that there are different organizational strategies for dealing with environmental pressures, such as acquiescing, compromising, avoiding, defying, and/or manipulating. According to Scott (1995), there are institutional environments at different levels, which determine the relationships between university, government, and external actors, while influencing and delimiting what strategies universities can use. Accordingly, different universities react differently to both external and internal pressures. Factors such as the history and cultural profile and roots, which a university develops in its early years, along with institutional environments and governance traditions, the complexity of institutions, and personalities of the main individual leaders in charge can be influential (Scott 1995). In this regard, to evaluate the

societal contribution of universities, the change in the university's institutional governance structure, the operational efficiencies, and the strategies for overcoming insufficiencies should be a part of an investigation. In addition, exploring the role of academics and their practices to resist change, negotiate, or alter the process to follow their own norms is critical.

*2.5. Conceptual Framework of Multilevel Governance Network*

Clark (1986) shed light on the complex dynamics of knowledge-based urban development, combining and integrating governance structures and organizational culture(s). Clark offered an analytical device, combining an approach privileging the influence of the context on the governance structure with a focus on intra and inter-organizational relationships that shape organizational actions and governance processes. In addition, he called for a shift in the unit of analysis, giving prominence to the university as an open system with multiple reciprocal relations and its own environment. Instead of using a holistic approach towards governance, he considered three different levels of governance: (1) the understructure, (2) the middle or enterprise structure, and (3) the superstructure. The understructure, shown by the color blue in Figure 1, considers the institutional structure of the focal organization, or university in this case, and focuses on intra-relationships. The middle or enterprise structure level, shown by the color gray in Figure 1, considers the inter-relationships of university and other regional and local actors that collaborate on city-regional development. The superstructure, shown by the color orange in Figure 1, focuses on the relationship of a university and its governing body, i.e., the government or Ministry of Education. The superstructure level considers the key role of government in underpinning and/or providing supportive regulatory environment and funding. Government policies and funding have both direct and indirect regional and municipal outcomes, which are both spatial and non-spatial (Smith and Bagchi-Sen 2010). In relation to the universities' contribution, the superstructure and understructure represent university governance, and the middle structure represents urban governance.

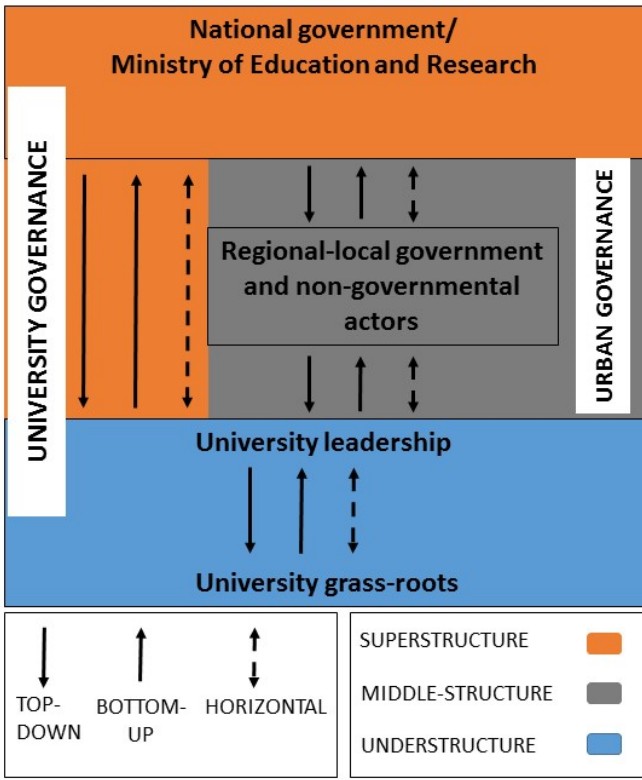

**Figure 1.** Different levels of governance structures, adopted from (Clark 1986).

Each level of governance can be characterized as bottom-up, top-down, or horizontal. In the 'bottom-up' type of system, the governing body follows, rather than leads, a change process initiated by the subordinate(s) within each level. Therefore, as Figure 1 represents, a bottom-up superstructure implies that the university, as the subordinate, has the autonomy, freedom, and control in planning and decision-making processes. At the understructure level, a bottom-up structure implies the dominant power or influence of the academics over the university leadership. At the middle-structure level, a bottom-up structure implies the dominant role, power, and interests of the community/university over the local-regional government. In the 'top-down' type of system, the subordinates merely respond to the superiors' suggested policy initiatives, who are enforced by their power and authority. In the horizontal type, neither superior nor subordinate dominates. They both have operational autonomy and the system, to a certain extent, is self-regulating. In the top-down or hierarchical version, reform processes are dominated by a closed group of top leaders, political or administrative/institutional, who score highly on both control and unambiguous organizational thinking, i.e., they know why and how to reform and tightly control the process. On the other hand, the horizontal version starts from the notion of heterogeneity in systems, institutions, interests, norms and values. In this version, "reform processes are more like a 'tug-of-war' between different actors, and decisions and implementation could be reached by majority, consensus, or sequential attention to goals and interests" (Christensen 2011, pp. 505–6).

## 3. Method

### 3.1. Methodology

The purpose of this qualitative research was to capture and understand the meaning of social-political actions and events that change over time. This requires learning and interpreting individuals' views and assessing a process of decision-making (Creswell 2008). When there is no pressure on the researcher to impose controls or to change circumstances (Denscombe 1998), the best research strategy is the case study that provides the opportunity to obtain an in-depth investigation of a given phenomenon within its context (Yin 2012). Trondheim could be an appropriate case to study governance in the university development process because the local government in Norway has historically performed crucial development functions and the government places great emphasis on governance, dialogue and cooperation between the state and cities, and between public and private parties (Regjeringen.no 2008). The case of the co-location of university campuses in Trondheim represents a typical case for testing the concept of governance in which a broad range of interests, strategies, conflicts, and power relations has been involved. Focusing on an unfolding chain of events and processes that change over time implies the longitudinal nature of the case study, which provides interactions within a context over time.

While exploring a particular process of the co-location and providing a rich history of its development is valuable for its own sake, it can shed light on problems and issues that may be common to other cases and can contribute to knowledge and theory building and help to refocus future investigations in the entire field.

The type of this single case study is 'embedded' (Yin 2012) and there are three levels of analysis: the understructure, middle structure, and superstructure. The main unit of analysis was the participants' discourses, while documentation and archival records were supplementary. This research should have looked for specific informants who have had important information and insight due to their positions or important actions (Table 1). Therefore, the sampling method was "nonprobability purposive expert sampling" (Trochim 2006), in which the researcher purposefully or intentionally selected individuals instead of conducting random sampling. To identify and get through to the key and most important actors for the interviews, documentation and archival records were the supportive methods. In addition, 'snowball sampling' was used as a complementary sampling technique by the identification of initial actors/interviewees who provided the names of other related actors and opened up possibilities for

an expanding web of contact and inquiry. A total of 25 interviews were taped, transcribed, coded, and analyzed through a 'Qualitative Data Analysis' tool. The interviewees (position + year) were respectively: 1. University professor and chairman of the committee for campus project (2004–2006), 2. University dean, Faculty of Social Sciences and Technology Management (2013–2016), 3. University project director (2004–2013), 4. University rector (2001–2005), 5. University professor, Department of Chemistry, 6. University internal board member in 2006, 7. University rector (2005–2013), 8. University director (2004–2006), 9. Adresseavisen journalist, 10. Municipal senior advisor, 11. University dean, Faculty of Architecture, 12. Student leader at student parliament, 13. University property administrator (2012), 14. Municipal chief executive (2000–2005), 15. Deputy minister of education (SV, Socialist Left Party) (2012–2013), 16. Deputy mayor (SV, Socialist Left Party), 17. University project manager (2009–2013), 18. University external board member in 2006, 19. University board leader (2005–2013), 20. Politician (AP, Labor Party) at Trondheim municipality (1999–2007) and later advisor at the Trøndelag county (2008–2015), 21. Minister of Education (2009–2013), 22. Politician (AP, Labor Party), 23. Board leader of NTNU2020 plan, 24. Minister of Education (SV, Socialist Left Party) (2005–2006), 25. Politician (SV, Socialist Left Party) at Trøndelag county in 2012.

**Table 1.** Decisive milestones in the history of the Norwegian University of Science and Technology (NTNU)'s co-location process.

| 1910 | 1922 | 1968 | 1996 | 2004 |
|---|---|---|---|---|
| Establishment of the Norwegian Institute of Technology (NTH) | Establishment of the Norwegian College of Teaching in Trondheim (NLHT) | Merger of NTH and NLHT, and creation of the University of Trondheim (UNiT), in which they had their own autonomy and separate rector | Establishment of NTNU by elimination of separate autonomies at UNiT. Introduction to the co-location idea | NTNU promoted the process of co-location further |
| **2005** | **2006** | **2006–2012** | **2012** | **2014–Present** |
| Change of management system at NTNU. A shift from the internal elected rector to the appointed rector | The case of co-location was stopped and rehabilitation of Dragvoll campus was brought up | Inaction on the rehabilitation of Dragvoll campus | The co-location idea was re-opened by the local politicians and the Ministry of Education took the full responsibility for the process | The co-location idea was approved by the government and today, it is in the planning phase |

Even though the Norwegian Center for Research Data (NSD) approved the non-anonymity of this research, the identities of the interviewees were kept anonymous and their quotations only include their role in this paper. The reason for this is that this research targets a wider society than the local one and non-anonymity is meaningless for (inter)national audiences.

*3.2. Introduction to the Case Study*

Trondheim is a municipality in Trøndelag County in Norway. It has a population of circa 200,000 (1 January 2018) and is the third most populous municipality but has the fourth largest urban area in Norway. Research and education have been the most important resources for the city's economic growth and development of the welfare society. Trondheim is regularly rated as the best student city in Norway, and students make up a fifth of the population (around 36,000). Trondheim is dominated by the Norwegian University of Science and Technology (NTNU), the Foundation for Scientific and Industrial Research (SINTEF), St. Olav's University Hospital, and other technology-oriented institutions, such as Equinor's Research Center. In Norway, the higher education system is centralized, with the Ministry

of Education and Research (in this paper, this is also referred to as the Ministry of Education or the ministry) having the overall responsibility for defining universities' rights and duties and financing, including grants for property development, which is acknowledged by the parliament.

Since Norway, as a participant in the Bologna process, is concerned about 'brain drain' from Norway to other countries in Europe, the future competitiveness of NTNU as the national technical university and Trondheim might have a significant impact on Norway's economic future. Therefore, while the social responsibility of education and research institutions has been taken into consideration, their interaction with the surrounding environment has increasingly been emphasized. Accordingly, Trondheim's ideal is to physically and institutionally integrate the university and college campuses into the urban fabric by combining the education and research functions with the city's activities and facilities. The location of the main campus of NTNU, Gløshaugen, the science and technology campus, centrally in the city, can enable them to meet their vision and integrate city and university. The Humanities and Social Sciences of NTNU are located on another campus, 'Dragvoll', which is 5.5 km from the city center and 3.5 km from the main campus Gløshaugen campus. Figure 2 (next page) shows the location of NTNU campuses in Trondheim.

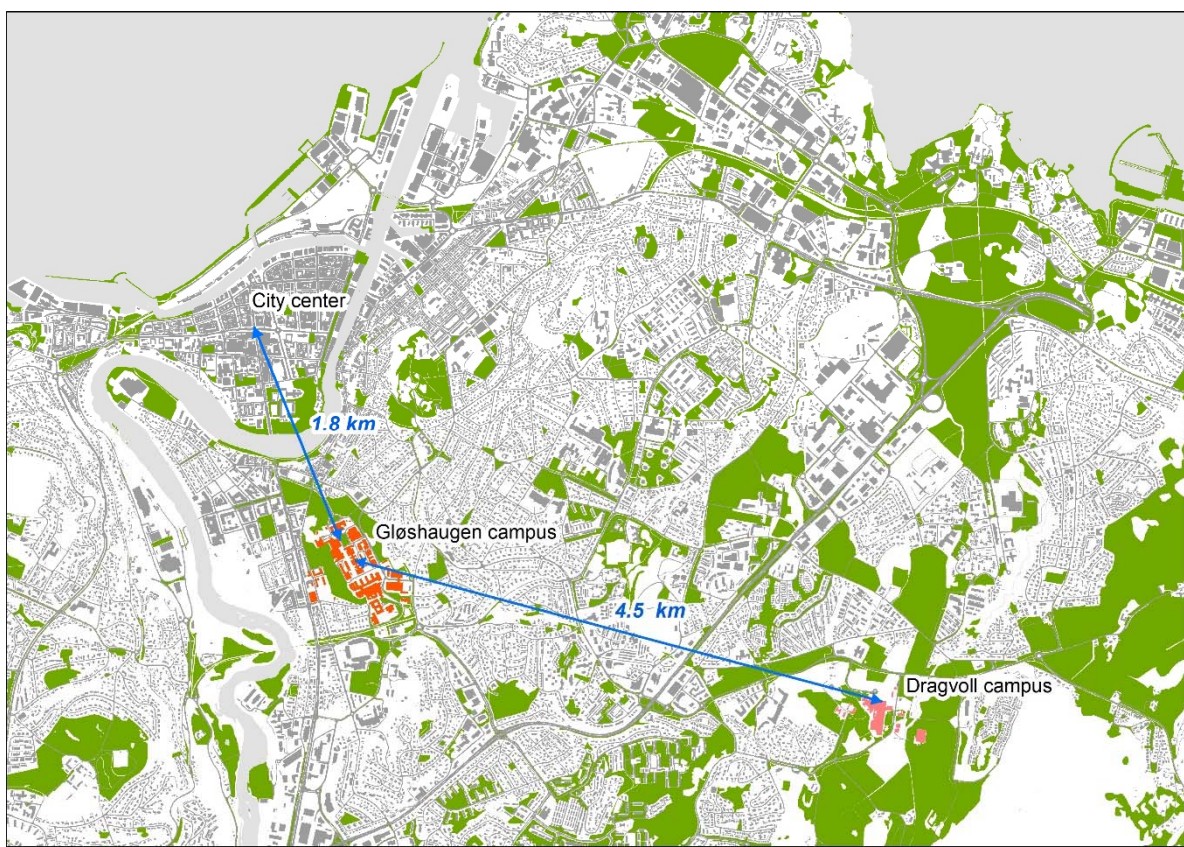

**Figure 2.** The map of the Norwegian University of Science and Technology (NTNU) campuses in Trondheim.

For many years, NTNU has considered the possibility of moving both the administration and education affairs of Dragvoll campus (80,000 m$^2$), to, or in close proximity to Gløshaugen Campus (Trondheim Kommune 2012). The assumption is that the re-location of Dragvoll would be more accessible for students to study different subjects from other faculties. Since many students live closer to the city center and Gløshaugen campus than to Dragvoll, such a location has been considered more effective and sustainable. This would lead to an efficient use of common educational areas, services, and facilities, while the university as a whole would become more accessible to the urban population. The Trondheim Municipality and the Trøndelag County have also supported and reinforced the

city-center-proximate solution for campuses, aiming at developing a hub of gravity for business, innovation, and other city activities (Trondheim Kommune 2012).

The collusion and simultaneity of assembling the activity of the two largest campuses (i.e., Gløshaugen and Dragvoll) with an ongoing discussion about the rehabilitation/future development of the Dragvoll campus, which has suffered from lack of space and facilities for students, forced the board of NTNU to choose a best option/strategy for the future in 2006. Despite the apparent supportive stances of various decisive actors, the board of NTNU came up with a 'no to the co-location idea' in 2006. Instead, the rehabilitation of Dragvoll campus, i.e., a two-campus model was prioritized, which was greeted with great surprise, and many supporters of the co-location were unhappy at this outcome. Apart from some minor changes, no important development/improvement happened at Dragvoll campus afterwards and after a period of inaction, the co-location of Gløshaugen and Dragvoll campuses was re-opened by regional and urban politicians in 2012. Then, the Ministry of Education took an intermediary role and directed the process of dialogue and resource-sharing, and was able to materialize the co-location of the campuses. The co-location process of campuses is now in the planning process and the Dragvoll campus is obliged to move its 14,000 students and 1500 employees to the Gløshaugen area.

The final outcome of the co-location case and the presence of positive insights and strong supportive actors to promote it at all times contradicted the dismissal of the case in 2006, the deviation towards the rehabilitation of Dragvoll campus, and the prolongation of the decision-making process. On the other hand, the board decision in 2006 (i.e., no to co-location) and the priority to rehabilitate Dragvoll campus controverted the following inaction period at Dragvoll (from 2006 to 2012) and their later agreement to co-location. Understanding the reasons and the contextual background for such an overturning was the basis of this paper. By focusing on the transformation of governance mechanisms, this paper investigated the transforming interactive capabilities of NTNU in response to the social and political dynamics behind the university's development. The transformation of governance structure at different levels over time can demonstrate the extent to which the societal contribution of university was/is impractical, inefficient, or misunderstood.

## 4. Findings

Table 1 above shows the milestones that were decisive in the development of the co-location decision. The word 'co-location' focuses mainly on the process of re-locating the Dragvoll campus to the Gløshaugen-Øya area.

### 4.1. The External Environment in the Case of Co-Location

During the economic recession in Norway around 1900, the demand for engineers and technological education increased to compensate for the economic downturn. Norway became an independent nation in 1905 and needed to have its own technical university on equal terms with the Royal Institute of Technology (KTH) in Stockholm and Chalmers in Gothenburg. In 1910, the Norwegian Institute of Technology (NTH) was established in Trondheim as an important element in the welfare state to meet the demand for an academically educated labor force within the country. According to the journalist of the local newspaper Adresseavisen, who has covered most of the campus development issues,

> "When they wanted to establish NTH in Trondheim, the issue was very disruptive and controversial for many politicians at the parliament, who preferred the national technology core in [the] capital city of Oslo, where there [is] more innovation, money, investment and people. Therefore, after the establishment of NTH, Trondheim became a very popular city. Many smart and clever people from all over the country started to move and live there."

The impact of NTH on the physical growth of Trondheim became significant, and industry and business in the region could begin to benefit from graduate engineers and architects. The story behind establishing NTH in Trondheim explains the culture and behavior of Gløshaugen (engineering) people

in the university development process. From the first day (in 1910), the NTH students were very conscious of their elite status (Jensen 2010).

According to the internal board member,

> *"The same sense of culture and community was true for the Norwegian College of Teaching in Trondheim (NLHT), established in 1922, and for the social scientists who today are located at Dragvoll campus. However, compared to NTH, it was gentle and less controversial and troublesome".*

In 1961, under the influence of the 'Ottosen reform' committee in Norway that aimed at integrating all post-secondary education into one higher education system, it was proposed that the Norwegian Institute of Technology (NTH) and the Norwegian College of Teaching in Trondheim (NLHT) be merged to create the University of Trondheim (UNiT). The aim was to make the education system more adaptive to student demand and the labor market. This was considered the first movement towards the decentralization of higher education, although the government was still supposed to control and steer the internal affairs of their institutions.

Due to the cultural differences between technical and social disciplines at NTH and NLHT, employees had little desire to be joined as a single institution. Eventually, it became the most complicated merger attempt in Norway, involving countless hours of opposition and negotiations (Opdahl 2004). However, in the end, the Ministry of Education bureaucratically decided to develop the University of Trondheim (UNiT) at Dragvoll farmlands to take account of the future expansion of the university and to encourage and enhance the city's development in that area (Eriksen 2007).

Despite the efforts made by the Ministry of Education to a create a stronger institution by merging NTH and NLHT, the UNiT was still a loose organization. The individual autonomy and different functions, backgrounds, and cultures backfired on the institutionalization of the UNiT and compelled them to reach an agreement on forming a new and permanent organizational structure for the entire university (Rabben 2016).

In 1988, the second reform of higher education in Norway, the 'Hernes Commission', aiming at the internationalization of higher education, was introduced (Mjøs 2000). A key element was the idea of a 'Network Norway', which meant that all institutions of higher education should be part of a unified system. In this regard, universities and university colleges that had previously followed separate laws had to become unified. As an effect of this reform, the organizational structure of UNiT was changed and the 'Norwegian University of Science and Technology (NTNU)' was created in 1996.

The government's decision to create NTNU in 1996 once again raised the university employees' opposition due to the latent conflict regarding their intrinsic professional differences. The employees' resistance was rejected by the government and the decision was again taken unilaterally and through a top-down process.

The government's coercion of the merger of the two traditions caused a period of estrangement and a deep long-running feud between the 'social' and 'technical' sciences which lasted for many years and influenced later processes. The social and technical faculties gradually lapsed back into their own traditions and followed their ways of implication in education separately and autonomously. This caused NTNU to suffer the lack of a sense of unity among their faculties, which had inhibited any positive collaboration and interdisciplinarity among their programs. The two traditions were more concerned about protecting their own areas of interest and competence than finding a progressive method of collaboration. According to the dean of the faculty of Social Sciences and Technology Management,

> *"There had always been some efforts to mix courses between the two campuses of Gløshaugen and Dragvoll. However, building a common culture by bringing them under the same umbrella was not an easy task. In addition, the physical distance was also a barrier that prevented their success. It was very challenging to move students up and down [between Dragvoll and Gløshaugen] all the time".*

The statement above indicates that the disciplinary division at the understructure level was influenced by their geographical distance. In this regard, the co-location of their campuses (geographical

proximity) was the underlying possible solution that the government raised to induce them to collaborate. The objective was multifaceted: creating larger and stronger disciplinary units, creating a common educational culture, enhancing contact and collaboration between different disciplines, offering students opportunities to combine subjects, creating better organizational conditions for adaptation to changes and societal needs, increasing cost efficiency, and improving library, information and communications technology (ICT), and administrative services.

In 2000, the third reform, the Mjøs Commission, proposed greater autonomy for the higher education institutions in Norway (Mjøs 2000). This reform asked for universities' and colleges' freedom to establish their own study programs without prior consent from the Ministry of Education, and for their independence in shaping their own future (Larsen 2002). In addition, it was determined that the ministry should appoint the governing body of the institutions (university board) with a majority of external representatives, which the rector/vice chancellor had to answer to. In this respect, the Ministry of Education proposed that Norwegian universities undergo a new management system with an appointed rector instead of an (internally) elected one and a new board in which half of the members should be external. NTNU was the first university which approved the proposal. Afterwards, the government decided to step back for a while and instead gave more freedom and autonomy to the university.

The transference of the autonomy to the university apparently represented a shift from a hierarchy to a more horizontal and negotiation-based relationship between the university and the government at the superstructure level. Subsequently, the former strong hierarchical model became more lenient (but still remained) and came under the shadow of the university's institutional autonomy. Such a minimization of government authority growth was also in line with the New Public Management (NPM) and neoliberal economic thoughts, which aimed at increasing the efficiency of universities by producing competition between them. The NPM idea was to separate the regulatory role and ownership of the state: indeed, to deregulate state monopolies to create competition (Christensen and Lægreid 2007). As a result, the political compromise was to give more freedom to universities to govern their own affairs, such as campus development.

Although the ministry's and NTNU administration's intention regarding the co-location idea might be allied, their strategy and the finance mechanism were in conflict in the new management system.

According to one of the professors,

> *"The NTNU administration, including the project manager and the university director, was doing too much on their own, especially on the things that they had no right or pure ownership to do, e.g., selling Dragvoll, financial calculations/pricings, and residential planning. The unilateral action of NTNU administration exacerbated the existing resistance, stubbornness, and controversial atmosphere within the university. Furthermore, the Ministry of Finance (FD) did not like NTNU's financial proposed model, which was based on a private-public partnership (PPP). The reason for this concern was that there were many other similar university development cases in Norway at that time, meaning that by accepting the PPP, the national government would be getting "a great deal of problems".*

As a result, due to the uniformity of the direction of the university and the ministry's dynamics and strategies towards co-location, the ministry removed its political and financial support, creating a large amount of uncertainty for the rector, the board leader, and others, who gradually diminished their interest in supporting the co-location idea. This gave an extra opportunity to the opponents, mainly the employees and the internal board members, to say no to the co-location in 2006.

After six years of inaction, according to the Minister of Education (2009–2013),

> *"In 2012 the co-location idea was re-opened by the local politicians because they were in need of developing the city and providing more residential areas. The campus at Dragvoll and the surrounding areas were attractive for that purpose".*

When the politicians (from the social leftist party/SV) re-opened the case of co-location in the city council in 2012, the Ministry of Education (from the same party) invited the university administration

and raised the co-location matter with them. In that meeting, the minister asked NTNU to stand aside and let the ministry take care of the case and carry out all the official and political procedures. This suggestion pleased the NTNU leadership because they would no longer have to worry about their financial solution. As a result, the NTNU leadership agreed, and the ministry took charge of the process by accepting full responsibility. Many believed that the ministry's accountability and new responsibility for the case of co-location was the only way that NTNU could get the money from the state/Ministry of Finance. The minister of education was the former Minister of Finance and had good knowledge about the political processes in the Ministry of Finance. Therefore, NTNU might not have succeeded in acquiring the money without the direct intervention and supervision of the Ministry of Education.

Since the 'no to co-location' in 2006, the central government's monitoring and control gradually returned and was enhanced again. According to the outcome of the year 2012, when the co-location idea was re-opened, the new superstructure type of governance was formed, which apparently aimed at an interactive problem-solving arena between the ministry and the university. The structural arrangement of governance has supposedly become heterogeneous, although it seems that the bureaucratic rationalism is dominant and 'solving on behalf of' is more evident than 'solving together'. Consequently, the current superstructure model stands somewhere between the traditional hierarchical (top-down) and the horizontal interactive forms. However, it is closer to the traditional mode in which the national government is a prime actor and plays a significant role among all stakeholders. Table 2 shows the process of governance change at the superstructure level.

**Table 2.** The superstructure governance model at different periods.

| Influential Event | Timeline | Outcome | Governance Model of Superstructure Level |
|---|---|---|---|
| "Ottosen reform" | 1968 | Establishment of UNiT (merger of social and technical disciplines) | Top-down hierarchical |
| "Hernes reform" | 1996 | Establishment of NTNU (removal of the individual autonomy and decision-making system) | Top-down hierarchical |
| "Mjøs reform" | 2000–2004 | More freedom and autonomy were given to the university | Still top-down, but university had more institutional autonomy in academic affair |
| "Mjøs" & NPM reforms | 2005 | Appointing external representatives to the university management system | Horizontal in appearance through delegation of power to the university board, but bureaucratic in practice |
| The campus development | Since 2006 | Gradually towards the reversal of traditional hierarchical/top-down model | Hierarchical but in the 'shadow of democracy'/mock democracy |

*4.2. The Urban Governance Environment in the Case of Co-Location*

In parallel with the organizational change for educational development, Trondheim Municipality also directed its strategies and attention towards a knowledge-based urban development. In 1997, the 'knowledge city' concept was launched as a new vision for Trondheim 2030 in the municipal long-term plan. Correspondingly, the university development had extensively mattered to Trondheim Municipality, who initiated a formal city-university collaboration. Knowledge development is/was not

an official and typical responsibility for municipalities in Norway, but due to Trondheim's uniqueness in this aspect, the municipality obliged itself to support it and acted as NTNU's complementary ally (Adresseavisen 2006).

Nonetheless, the involvement of the municipality, in this case, at the beginning, was interpreted as a great interference that led to skepticism and confusion among different actors, both at NTNU and in the municipality. Employees at NTNU felt that the municipality or someone outside NTNU was leading this case and that they were obliged (and dictated to from above) and their dissenting voice (the democracy) was not taken into account. One of the university professors mentioned that

> Because of the interest the municipality showed for the Dragvoll properties, many actors were suspicious that the municipality had other intentions and was more interested in the properties at Dragvoll than helping NTNU, for instance, building homes for elderly people.

Trondheim Municipality has been the strongest supporter that NTNU has had in the whole process. The municipality gave full backing from the beginning, especially regarding the land-use planning (for Dragvoll property) and adapting the local development plan in the Gløshaugen area to meet NTNU's needs. This shows how the middle and understructure levels can influence each other's functionality.

The urban or middle-structure governance model, in which the university has interacted with other local and regional actors in this case, has been unchanged and constantly been horizontal and self-organized, implying complex reciprocal interdependencies.

Neither university governance (at superstructure and understructure) nor urban governance has undergone a structural change due to each other's influence only affecting their quality of relationships. At the beginning, when NTNU was more autonomous in the procedure of its development, its relationships with other local/regional partners were collaborative, negotiated and mutual-benefit-based. Both the city and the university administrators and leaders had based their collaboration/negotiation on the fact that NTNU is good for the city and the city is good for NTNU.

However, the employees and leadership of the university had an inconsistent perspective and interest regarding the involvement of the municipality, which caused many internal tensions and conflicts. Consequently, the strong network between the university and the city authorities at the middle-structure level resulted in a significant reduction in high-trust relations within the understructure governance. The conflict between the employees versus the city and university authorities had undermined the effectiveness of the governance at the understructure level, which was also associated with the 'no to co-location' decision in 2006.

On the other hand, the financial requirement for actualizing the co-location of campuses had created an illusory interdependency and resource-exchange relationship between the university and other local partners, which was inconsistent with the ministry's political strategy. The strong collaboration between the university leadership and the city made the university misunderstand its real dependency on the government, which undermined the quality of superstructure interrelations as well. Thus, the horizontal self-governance model that somehow existed at all levels of governance, but at different scales, resulted in a lack of transparency and accountability due to the formation of informal and interpersonal networks, particularly at the level of the individual, rather than at the collective level.

The 'no to co-location' decision in 2006 and later, the government's re-initiation of co-location in 2012, caused the university to realize the ministry's real power and significance in the decision-making process. Thus, the university has improved its collaboration and communication with the ministry. However, the university's relationship with the municipality and other local-regional partners has worsened. Today, the shared desire of the university and the ministry to create a single campus to boost international competitiveness has been more highly emphasized, becoming at odds with the city and region's interest in having the university campuses more integrated into the fabric of the town. Thus, the city and university authorities are struggling to reach a compromise on the co-location site. The municipality and the university both want to co-locate campuses near the city center (northern part of

Gløshaugen campus). However, they are in conflict over building in the park, which the university is interested in, but the municipality and politicians are against (due to environmental issues and the protection of neighbors' interests).

Due to the financial barriers and availability of land, the university's present option is to build in the southern part of Gløshaugen campus, which makes the availability of land less problematic. Accordingly, the enhancement of the relationships at the superstructure level, based on 'solving together', has evidently undermined the dependency of the university on its local partners and vitiated the quality of middle-structure relationships. On the other hand, it seems that the relationship or perspective of employees towards the role of the municipality has unexpectedly and completely reversed since 2006. Today, the impression is that the elected representatives of employees on the board and other academics, particularly planners and architects, align themselves with the municipality's strategies regarding NTNU's campus project. This explains how the change or evolution of actors' interests/alliances over time can influence the governance mechanisms and interactions at different levels. Therefore, as Potts et al. (2015) argued, any analysis of governance underpinning complex systems must consider how the system is structured and organized, as well as the way in which the structures in the system function. While 'structure' implies the way, actors stand in a network and interact with each other, 'function' implies the way different components (such as power, agency, and the networks of actors) influence and shape actors' actions/decisions over time.

Thus, exploring both structures and functions can enable us to take a more systemic view of the university governance system while still accounting [in a non-linear way] for the numerous dynamic interactions of multiple structures across scales and policy spheres (Potts et al. 2015, p. 13). Table 3 shows how the university's interactions at one governance level influences its relations with the other levels.

**Table 3.** Governance interactions at the middle-structure level.

| Influential Event | Quality of Relationships as a Result of Interaction between Levels | Governance Model of Middle-Structure Level |
|---|---|---|
| Before 'no to co-location in 2006' | The strong network between the university and the city authorities (at the middle-structure level) had resulted in the significant reduction in high-trust relations within the understructure governance | Horizontal and self-governance |
| After the 'no to co-location' in 2006 | The strong network between the university and the ministry (at the superstructure level) has weakened the relationship between the university administration and the municipality | |

### 4.3. NTNU's Internal Governance in the Case of Co-Location

The government's change of strategy at the superstructure level, under the Mjøs reform in 2000, also influenced the understructure (internal) governance model. Successively, in the early 2000s, the governance model at the understructure level became quite bottom-up, meaning academics/employees had the power to collectively make decisions about their academic affairs. Therefore, when the idea of co-location came to the force (in 2004), the university administration, including the university director and the rector, hired a project manager and seized the control of and responsibility for campus development and planning. The change of understructure governance model might be a result of the apparent shift of superstructure governance to a more horizontal form.

Later, in 2005, increasing global pressure and an emphasis on universities' international research excellence and capacity for strategic management in pursuance of NPM on the one hand, and on universities' contribution to knowledge-based urban development on the other, imposed a new management system on NTNU. The rector was no longer elected; instead, he was appointed by the board, of which half the members were appointed by the Ministry of Education. Subsequently, the board became the highest governing body at NTNU and the university director, who was one of the main actors in the last round, gradually lost part of his positional power in the new system and left the university. The new management system resulted in the inevitable involvement of multiple actors in the campus and university development, facilitated a shift in NTNU's role and relationships with other local institutions, and induced 'entrepreneurial competition' on NTNU's agenda to be more efficient and responsive to outside demands. The financial model that the university administration came up with for the co-location to sell Dragvoll properties in collaboration with the municipality and other potential private partners verifies such a shift towards a market-based norm.

The change of management in 2005 eased the resistance process for the opponents for two reasons. First, the university director, as one of the strongest actors and main barriers, lost his power/influence. Secondly, a new rector and board leader came onto the board and he had a moderate and peaceful attitude and preferred to inhibit any disagreement and act as a mediator. These two reasons became advantageous for the opponents. The board leader's moderate strategy made the external board members, who were usually in alignment with the board leader, give up their supportive votes. In addition, due to the newness of the understructure governance model, the external members did not yet directly interfere in the university affairs at that time. On the contrary, they abided by democracy and protected academic freedom from external interests. This situation gave extra power, in terms of opportunity, to the internal board members, who were the most strong-minded opponents. As a result, the internal board members were able to counteract the power of the university leadership and administration and leverage their demand extremely effectively.

On the other hand, the amount of money that would be gained by selling Dragvoll properties could not be estimated precisely due to the future market uncertainties. In addition, NTNU (similarly to any Norwegian university) did not own the university property and thus did not have the power or the right to decide upon the financial resources and build the campus completely on their own. Moreover, the private-public partnership was not the model that the finance minister and many politicians of different parties wanted to support. Therefore, while the governance model at both superstructure and understructure was changed due to NPM reform, market competition or partnership with the private sector was not an optimal financial model for the national government at that time. Nonetheless, since NTNU was/is a state-owned organization, the public fund was another considerable source/solution that NTNU could take into account. However, in order to obtain public funding, NTNU had to stand in the queue with other national projects, making it difficult to predict when it could proceed. Thus, the level of uncertainty for this alternative was also very high, unless they had a governmental guarantee of financial support. According to many interviewees, in addition to the traditional conflict between social and technical sciences that made employees resistant to the co-location decision, the lack of the national governments' political and financial support temporarily dissolved the case of co-location in 2006. Due to the inappropriateness of the facilities at Dragvoll campus, the board of NTNU brought up the rehabilitation of Dragvoll.

The internal board member's (from Dragvoll) statement below partly explained the inaction on the rehabilitation of Dragvoll campus afterward.

*"Since the no to co-location decision in 2006, NTNU has had the rehabilitation of Dragvoll at the top of its priority [list]. Despite numerous meetings with the government agencies, rehabilitation of Dragvoll was not realized. This proves that NTNU has become a part of a political game, where its desire for further development at Dragvoll conflicted with the local and regional politicians' ambitions for urban development. As a result, the Ministry of Education (KD) systematically neglected NTNU's*

*wish on a two-campus solution. We [the board of NTNU] greatly regretted our lack of action that has deteriorated the learning environment at Dragvoll and NTNU".*

According to the events that happened after 'the no to co-location' in 2006, it was concluded that the change of management system in 2005 conversely imposed the university to more scrutiny, control systems, and a hierarchical mode. In fact, the co-location idea was stopped due to the realization of a democratic decision-making process at the university where employees had great voices. However, the later understructure governance development apparently thwarted the democratic participation and encouraged passivity within the university community.

Since the idea of co-location was initiated in 2012, the external board members have made the strategic choices, while the rector, deans, and chairs of departments (the internal members) are apparently responsible for ensuring that their strategies are implemented. In a way, the new internal governance reflects a line of hierarchical command which goes from the board and the rector through the deans and to the chairs of departments (Rasmussen 2015). In addition, today's internal situation looks like tokenism, in which employees can both hear and have a voice, but they do not have the power to ensure whether their views are heeded by the leadership/authority (shadow of democracy/mock democracy). Therefore, they cannot follow through with what they really mean/want and the right to decide is still limited to the upper authorities. In addition, the academics' engagement in the day-to-day administrative life of the university was withdrawn, which has produced an organizational separation between employees and managers/administrators (Rasmussen 2015). Table 4 shows the change of internal governance at the understructure level over time.

**Table 4.** Understructure governance model at different periods.

| Influential Event | Timeline | Outcome | Governance Model at Understructure Level |
|---|---|---|---|
| Ottosen & Hernes reforms | 2000–2005 | More freedom and autonomy were given to the university. The rector was elected | Bottom-up professionally-based |
| Mjøs & NPM reforms | 2005 | Appointing external representatives to the university management system. The rector was appointed | Horizontal and self-governance |
| The campus development | Since 2006 | Gradually shifting towards the reversal of a traditional hierarchical/top-down model | Hierarchical but in the shadow of democracy |

## 5. Analysis

*How the Multi-Level Governance Networks Influenced the University Development Processes and Its Societal Contribution*

In this section, the interactions of network governance at three levels—the understructure, middle structure, and superstructure—are analyzed to answer the research question. With the reference to Oliver (1991) theory (explained in Section 2.4), in the initial phase of the co-location, the ministry had the only autonomous administrative power and the superstructure governance was evidently top-down. The government's unilateral decision on establishing the UNiT and later NTNU, regardless of employees' internal opposition and opinion, was a great affirmation of the real hierarchy and the bureaucratically-based structure. The government's decision to merge the technical and social institutions (NTH and NLHT) was taken through the manipulation of agreement (Sager 1999). The

initial agreement was that each institution had its own autonomy. However, their autonomy was gradually taken from them. The manipulation consisted of hushing up the reality of planning and decision-making procedures, particularly in terms of time and financial factors. By withholding the useful and complete information, the government deliberately tried to influence the response of the academics/university administration with the tacit intention of supporting a particular alternative (first institutional and then physical merger). Therefore, the academics' desired response was not explicitly communicated. In this regard, the obscure message about merging the institutions was concealed from academics and this rhetoric tacitly promoted a desired interpretation.

On the other hand, the employees' reactions to the forced merger were in the form of surrender, lack of resistance, and tolerance of the situation, mainly because they had no other choice, due to their asymmetrical (formal) power versus the government. It can be concluded that the internal university resolution strategies result from their formal power, which stems from the structural model of governance.

The lack of employees' sense of joint responsibility for and ownership of the decisions, deliberately or not, hampered the implementation of university institutional development, rather than supporting it. The later disunity and disconnection between university traditions were the signs of the growing disaffection towards the government's one-sided decision. For that matter, the government's victory was not complete and the positive interaction and collaboration that the ministry aimed for by bringing the two traditions under the same organization was not effectively achieved.

As time passed, referring to the no to co-location in 2006, it became evident that although the government had the formal power/authority to make a unilateral decision, academics also had a relatively strong mindset and bargaining power (informal power) to gradually influence or manipulate the outcome. As a result, the bureaucratic and hierarchical governing model may not have been productive at that time (1996), which urged the government to give more power/freedom and autonomy to the university.

The introduction of the new university governance change was always quite piecemeal, which made a reversal of the power less apparent and the decision-making process less transparent for the involved actors, particularly the university (Rasmussen 2015). At the understructure level, such an ambiguity caused a dichotomy of strategies for dealing with the external environmental pressures. On the one hand, the strategy of the rector and the board leader was in the form of conformity to institutional pressures (acquiescence) that involved balancing, pacifying, and bargaining with others (compromise). On the other hand, the university director and the project manager attempted to disguise the internal and external nonconformity, buffered their endeavors from institutional pressures, and somehow escaped from institutional rules and expectations, i.e., tacit bargaining or outmaneuvering (regarding the financial model they proposed). Employees had yet another strategy, which was to resist internal and external pressures in a very public manner. They used different tactics of defiance, including dismissal, challenges, and attack.

In connection with superstructure governance, the government's rejection of a public-private partnership and the prevention of the full market regulation or some part of it showed that the government was just allowing for some degree of institutional autonomy at NTNU, but it was keeping a firm hand on the regulation of the university system, making only rhetorical use of the 'market'. As a result, when the ministry saw that NTNU administration was unwittingly proceeding with the case through a self-steering process, it tried to obfuscate and misrepresent the financial possibilities, acting as a gatekeeper, and backed off from the necessary support. The lack of governmental financial and legal support created a great amount of uncertainty for the university leadership that influenced their support for the co-location idea. Accordingly, the government's strategy to control such a conflicting environment was 'manipulation' and 'false assurance' (Forester 1989), proving that the ownership and jurisdiction of a university is/was conditional and limited by the ministry as the main owner. As a result, although the idea behind the self-organization model was to provide opportunities for several voices to be heard, it created a misalignment of leadership, power and role: first, within NTNU

internally and, second, between NTNU and the ministry. Even though the university governance had been changed in 2005, in fact, the government's centralized and detailed control over the university's institutional autonomy remained and partly delegated to the university board, in which half of the members were appointed by the government. As a result, it was not easy to locate the locus of power and identify where decisions were taken and who was responsible for the 'no to co-location' decision in 2006. In addition, the absence of some sort of authority and legal power rendered the decision situations unstructured, anarchic, and less predictable so that a variety of influences were brought to bear on choices.

On the other hand, it might be problematic if the ministry suddenly and entirely re-changed the embedded norm of NTNU in which the university had just recently gained a large degree of discretion for dealing with its own planning processes, based on internal consensus. Although the governance understructure was converted from a professionally-based bottom-up model to a more collaborative and negotiation-based governance model in 2005, the change was very recent, and employees still had greater discretion over university affairs. Subsequently, the ministry obfuscated the planning and decision-making procedures and, by rejecting some of the financial decisions that NTNU made, tried to express its superior power and role. The statement of the former minister of education (2000–2002) below depicts the formalization of the relationship between the Ministry of Education and the university board, which had faced a major shift during that time.

> "If the minister of education of the time (2005–2006) had been upfront about his intention and given a clear mandate for the co-location alternative, the university board had to follow it up. However, the minister did not offer a clear point of view and thus the board leader was free to say what she thought".

The university rector (2000–2004) had a similar opinion, stating:

> "At that time, NTNU had an annual budget of 3 billion kroner. The cost of moving Dragvoll down to Gløshaugen was estimated to be between 100 to 400 million kroner, which was much less than the annual budget. Therefore, it was not a question of political money, it was a case of political will. If the government wanted to support or had trust in NTNU's capability, then even the 400 million would be a quite reasonable price".

The period between 2006 and 2012, the inaction on the rehabilitation of Dragvoll campus, was an adaptation and a learning process towards governance transformation, not only for NTNU but also for politicians from all levels and for the government. During this period, they gained more insight into the reality and substance of power relations, leadership, and conflict resolution (how governance functions) to realize which governance structure/model is practical and efficient for their own or for the collective advantage. In addition, *time* was necessary for different actors to adapt to the recent governance structural changes, which required flexibility and the modification of preferences and strategies. The government used this period to re-balance/heal the university governance relationships. Meanwhile, by making some inexpensive and unserious developments at Dragvoll campus, they tried to silence the opposition, making the condition optimal for bringing up the case of co-location again. Thereby, the traditional university democracy, in which the university had quite a lot of autonomy, met its end (Rasmussen 2015) in this period (2006–2012), and the central government's monitoring and control was returned and enhanced again. According to the former minister of education (2000–2002):

> "The rector saw that there was no political support for rehabilitation of Dragvoll. The rector needed political support to push through a new building at Dragvoll, but he realized that he could not do this alone in the long run, when the mayor, the county mayor, and the former minister of education stood against him. This caused the rector to be open to a new investigation of co-location again".

The statement above, especially from a governmental actor, shows that such an outcome was more influenced by the informal interactions of actors outside their formal relationship that encourage

actors to readily engage in tactical ploys of direct action, implying realities of distortions, politics, and power plays (Hillier 2000). Accordingly, many actors in formal structures and relationships, e.g., the university, may be unable to grasp the political sensitivities and ambiance built during the process and realize the real perspectives and interests of involved actors. In addition, the longitudinal framework of this research shows that in such (organized) anarchy, actors' preferences/interests are discovered only through actions, implying the importance of temporal factor as well.

Another aspect of the temporal factor was explained by many interviewees, who believed that time was needed and that NTNU people, who had the tradition of living in an 'ivory tower', were conservative and impervious to any change. Thus, they needed time to realize that their one-sided negotiation processes would not only cost them time, energy, and money, but also would prevent them from recognizing the power and influence of the national and regional governmental and political actors. According to the rector (2002–2004):

> *"As more time went by, many opponents were convinced that the governmental decision on establishing NTNU was not that bad, because at that time NTH was very conservative and arrogant. The nostalgia connected with the unique institution of NTH made them stubborn and closed-minded to accept any change or improvement. Thus, it was necessary that the government took that decision unilaterally!"*

Even though the co-location case could finally silence the NTNU's opposition and satisfy many actors' interests in 2012, today's NTNU and city-regional parties are challenged to compromise on a co-location site. It can be concluded that developing a mutual beneficial outcome and managing contradictions are not things that are done once and forgotten about; they are continuous processes. This also verifies the importance of a temporal factor for studying governance.

## 6. Conclusions

The underlying assumption of this paper was that 'governance', which is based on the negotiation and collaboration rationality between universities and external actors, can help universities to balance internal and external pressures and provide a better societal contribution. The question was: 'how do the multi-level governance networks influence the university development processes and its societal contribution?'

The decisions upon university development have different dimensions that touch upon various types of policies, government levels and actors, whose interactions determine the course and content of the development process. Actors are often involved in more than one policy arena and their interests, perceptions, and strategies are contrasting, unpredictable, and subject to change. Therefore, the decision-making system concerning university development is both fragmented and complex and can be challenged by different factors, such as the presence of unresolved tensions/conflicts, e.g., disciplinary conflicts between social and technical faculties of NTNU, the cultural profile that the university has developed in its early years, the complexity of universities' institutions and personalities in academia, the lack of government financial and political support, uncertainty about access to the critical resources, distortion and changes of the policy process, a secret political collusive relationship within the government which the outside world could rarely glimpse, and a misalignment of strategies and power among main actors. Therefore, handling university development is more than a simple process of negotiation and making a compromise. The fragmented and multi-level nature of the governance network is another source of complexity.

Investigating the impact of a multi-level governance network on university development and societal contribution in the case of NTNU in Trondheim has showed that superstructure and understructure levels (university governance) are more interconnected and interdependent than the middle structure (urban governance). The government has been the one who determines the institutionalization of understructure and superstructure levels, either in response to internationalization or in reaction to the mechanism and practices of the understructure governance. If the government is not satisfied with the way the university autonomously aligns and adjusts itself to

the institutional changes and other endogenous and exogenous events and pressures or the way it generally functions, a modification of university governance usually takes place. Such a modification can be slight or significant and gradual or sharp. Accordingly, the university has little or no control over its inevitable institutional changes.

Governance relationships may become fractured and need healing, repairing, or replacement, which requires the government's intervention. It is not always practical for the government to steer from a distance, providing a level of autonomy or mediating its control. Sometimes, this scenario requires gaining or maintaining political power. Therefore, depending on different situations and the prerequisite capability of governance for actors at different levels, e.g., resistance of the university's community or the university's autonomy in decision-making in the case of the co-location in Trondheim, the government should act differently.

The transformation of the governance structures and subsequent power struggles or resistances might have been rooted in the background of a university when it was established. Therefore, it is important to understand that the processes of organizational and institutional governance change, including how and why they happened and whether the change was in response to an external crisis, e.g., uncertainty in government funding, a change of the national mood, or an internal crisis, e.g., lack of confidence in leadership or organizational resistance. We conclude that the balance between the democratic governance problems and potentials depends on the institutional conditions, power struggles, and national governmental policies/priorities. In addition, exploring governance requires political, cultural, and periodic review. This also shows that planning and decision-making have unpredictable developments and that reaching the desired societal contribution is not dependent on a single factor or specific course of action; indeed, several factors/courses of actions should coexist and coincide. As a result, instead of taking a holistic approach towards governance, attention should be paid to the interaction patterns at different levels and the ways in which individual actors and organizations fit together.

In the case of co-location, the government had to give the university some freedom and autonomy to reduce existing internal conflict, which was rooted in the time when NTNU was established, an outcome which manifested as a 'loose coupling' of different disciplines (UNiT) that created a heterogeneous set of norms and subcultures and inconsistency of interests within the system. Afterwards, by letting the time pass, the government regained its power. There was also an interim opportunity for collective learning as well as the establishment of shared visions and trust among diverse stakeholders. Therefore, different actors might have had a common interest in university development, but misalignment of their dynamics and strategies could challenge the efficiency of the process. In such situations, we believe that the government or the Ministry of Education, in similar cases, can take the intermediate role and direct the process of dialogue and resource-sharing in university development procedures. In order to mediate the failure of governance and facilitate the university development processes, the government has the power to reintroduce hierarchy into the equation. The paper shows that although the government/hierarchical mode of governing can also fail, the government's initiative and control functions or some moderate hierarchical coordination are still necessary.

This paper does not suggest 'the government's control functions' as a clear-cut recipe for dealing with university governance challenges. On the other hand, we tried to elaborate on the theoretical and empirically founded insights of the governance approach and provided a recommendation for dealing with the potential deficiencies that can undermine universities' societal contributions. We are aware that our ideas require further elaboration and that they will not always be practical in other cases, and there may be another solution that we have not foreseen. Further research is necessary to extend the findings of this research and to increase the lessons in policy and practice for stimulating multi-level governance processes and structures in other contexts, and in relation to knowledge-based urban development.

**Author Contributions:** Conceptualization, S.G.; Formal analysis, S.G.; Methodology, S.G.; Supervision, T.M. and R.A.; Writing—original draft, S.G.; Writing—review & editing, T.M. and R.A.

**Funding:** This research received no external funding.

**Acknowledgments:** We thank Carmel Margaret Lindkvist for her fruitful comments and Yngve Karl Frøyen for providing the GIS map. We would like to express our special appreciation and thanks to the reviewers and the academic editor for the valuable guidance, substantive time and inputs that they have made, and their contributions to raising the overall quality of the paper.

**Conflicts of Interest:** The authors declare no conflict of interest.

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
