# Peer review of "Exploring the Impact of Complex Multi-Level Governance Structures on the Societal Contribution of Universities to Knowledge-Based Urban Development"

_socsci, doi:10.3390/socsci8100279_

Round 1

Reviewer 1 Report

The central challenge that I have with this paper is that I found it difficult to follow due to a lack of fluency in English writing, and organizational issues. Extensive English language editing is required for the paper.  There are frequent mismatches between tenses and there is a tendency to place articles in front of nouns that is incorrect. (E.g University governance does not take an article). Sentences are unnecessarily convoluted and there is an over-reliance on direct quotes, that makes the paper rather stilted and hard to follow.

Substantively, I found the argument of the paper potentially interesting and really worth re-packaging in a more polished form. There are a few issues that the authors should address: 

First, the authors need to be attentive to not making broad claims about historical context. For example: "Before, universities were designed/seen as closed community, 100 ivory towers, in which the academic community needed to be kept away from unwanted distractions," (lines 99-100) needs both specificity in terms of the when and where of "before," in addition to citations to substantiate the claim. 

Second, and more generally, there needs to be some attention to what  geographic context is being discussed.  There is no monolithic system of university governance in Europe, so the author(s) need to be careful when making claims of "in Europe." 

Third, the section on University governance is confusing to  me. There appears to be a conflation of the history of state-funded public universities with the concept of university governance. I think what might be needed here is a clearer separation of the issue, with a separate discussion of the development in Europe of the state funded public university, then a discussion of the development of university governance models. Secondly, rather than the reference to faculty grass-roots, there should be a discussion of peer governance (collegial governance)  which is an important element of university governance systems, including its interactions with state actors. 

Fourth, there is a challenge with the organization of the paper. The methods section comes quite late in the paper; as a result it is a bit of challenge to link the analytical discussion with the literature review that comes before. 

A minor point of clarification --what is the reference (see Chapter 2) in lines 168-169 referring to? 

Author Response

Response to Reviewer’s Comments

Dear Reviewer,

Coauthors and I very much appreciated the critical, constructive and positive comments on this manuscript by the reviewers. The comments have been very thorough and useful in improving the manuscript and increasing the scientific value. We have taken them fully into account in revision. The manuscript has also undergone English language editing by MDPI.

Our responses to your comments are as follows:

Reviewer #1:

The central challenge that I have with this paper is that I found it difficult to follow due to a lack of fluency in English writing, and organizational issues. Extensive English language editing is required for the paper. There are frequent mismatches between tenses and there is a tendency to place articles in front of nouns that is incorrect. (E.g University governance does not take an article). Sentences are unnecessarily convoluted and there is an over-reliance on direct quotes, that makes the paper rather stilted and hard to follow.

Response- Thank you so much for your comment. We used MDPI English editing service to check the text for correct use of grammar and common technical terms, and to edit it to a level suitable for reporting research in a scholarly journal.

The authors need to be attentive to not making broad claims about historical context. For example: "Before, universities were designed/seen as closed community, 100 ivory towers, in which the academic community needed to be kept away from unwanted distractions," (lines 99-100) needs both specificity in terms of the when and where of "before," in addition to citations to substantiate the claim.

Response- We have taken reviewer’s comment in full consideration and have avoided general claims. for example: We have replaced the statement above with: “Until a few decades ago, many universities were seen as closed community or ‘ivory towers’ in which the academic community needed to be kept away from unwanted distractions (Amaral and Magalhaes 2002; Barrett 1998). According to Barrett 1998, Thirty years ago, universities catered for a relatively small sector of the population. or according to Amaral and Magalhaes 2002, “until a few decades ago, the prime feature of the university was not its relevance to society, so much as its detachment from society”

there needs to be some attention to what geographic context is being discussed. There is no monolithic system of university governance in Europe, so the author(s) need to be careful when making claims of "in Europe."

Response- We have taken reviewer’s comment in full consideration and added supplementary citations.

the section on University governance is confusing to me. There appears to be a conflation of the history of state-funded public universities with the concept of university governance. I think what might be needed here is a clearer separation of the issue, with a separate discussion of the development in Europe of the state funded public university, then a discussion of the development of university governance models. Secondly, rather than the reference to faculty grass-roots, there should be a discussion of peer governance (collegial governance) which is an important element of university governance systems, including its interactions with state actors.

Response- We have added a section: ‘From ‘State’ and ‘Market’ to ‘Governance’ to clarify what we mean by ‘governance’. Then we wrote about the similar transformation processes within the higher education governing system. Hopefully now you will find it rational.

there is a challenge with the organization of the paper. The methods section comes quite late in the paper; as a result, it is a bit of challenge to link the analytical discussion with the literature review that comes before.

Response- We have taken the editor’s suggestion on the re-organization of the paper and it will be well reflected by the revised version of manuscript.

A minor point of clarification --what is the reference (see Chapter 2) in lines 168-169 referring to?

Response- Sorry for the error. We have deleted it.

Reviewer 2 Report

This article is based on a case-study in Trondheim. It looks at the decisions, and controversies, about the co-location of two separate campuses of the Norwegian University of Science and Technology, previously separate institutions with different histories, values and traditions, within the framework of a discussion of multi-layered governance - the national layer (superstructure), the regional or civic layer (middle or ‘enterprise’ governance) and the institutional layer (understructure). This discussion is itself framed within the wider context of the evolution of universities in terms of urban development (clever / science cities)  and generally (stats steering, administrative devolution, marketisation and the New Public Management etc).

It addresses an interesting and important topic. But it has a number of weaknesses, both theoretical and methodological, which would need to be addressed before it could be published.

Theory and conceptual structure

A number of important, and well-used, concepts are employed. These include ‘creative destruction’ and the ‘New Public Management’. I am not convinced these have been used sensibly or accurately. For example, the author uses ‘creative destruction’ to describe the role of knowledge institutions (such as universities) in the creation of new - and destruction of old - knowledge, in contrast to the destruction and creationism of capital through the decay and formation of companies, even though this was its original use. Similarly NPM is used as virtual synonym for privatisation and marketisation, although its original and more accurate use was with regard to the reform and modernisation of public institutions through the application of business techniques. The term ‘postmodern production’ is used, the meaning of which is unclear. What is missing is a clear exposition of theories and concepts of governance, which can then be used to illuminate the case study. As it is, the discussion of concepts and theories is only very loosely related to the discussion of the empirical findings.

Methodology and case-study

The article, derived from a PhD thesis, has two weaknesses in my opinion. First, it lacks of clear description of (a) the university and its campuses; and (b) the actual processes of decision making on the issue of their co-location. As a result it is difficult for the reader to get a clear picture. For example, a simple chronology of decisions by the various actors would have helped. Secondly, the article is based on interviews with key actors. Although this is a decent number, no indication of how many interviews were conducted with which types of actors (and the basis for selecting interviews). Also almost no narrative - direct quotations or summaries of views expressed - is given. Instead there is a series of general comments / conclusions.

Rather too much of the quoted academic literature is also rather dated.

Conclusion

The overall conclusion is unclear, apart from a restatement of the research question (the respective influences of different governance levels). Little attempt is made to indicate what contribution has been made to new knowledge, either in a theoretical sense or in terms of public policy making.

To some degree these weaknesses arise from the, at times, poor English in which the article is written. It does not matter that odd words or missing or that the grammar is incorrect; this could easily be remedied. More serious is that because of the way in which some ideas are expressed the points being made and the argument developed is sometimes obscure and difficult to follow.

I have reluctantly (because it is an interesting topic and the author has clearly worked hard) come to the conclusion that it is not suitable for publication in anything like its current form. 

Author Response

Response to Reviewer’s Comments

Dear Reviewer,

Coauthors and I very much appreciated the critical, constructive and positive comments on this manuscript by the reviewers. The comments have been very thorough and useful in improving the manuscript and increasing the scientific value. We have taken them fully into account in revision. The manuscript has also undergone English language editing by MDPI.

Our responses to your comments are as follows:

Reviewer #2:

Theory and conceptual structure

A number of important, and well-used, concepts are employed. These include ‘creative destruction’ and the ‘New Public Management’. I am not convinced these have been used sensibly or accurately. For example, the author uses ‘creative destruction’ to describe the role of knowledge institutions (such as universities) in the creation of new - and destruction of old - knowledge, in contrast to the destruction and creationism of capital through the decay and formation of companies, even though this was its original use. Similarly NPM is used as virtual synonym for privatisation and marketisation, although its original and more accurate use was with regard to the reform and modernisation of public institutions through the application of business techniques. The term ‘postmodern production’ is used, the meaning of which is unclear. What is missing is a clear exposition of theories and concepts of governance, which can then be used to illuminate the case study. As it is, the discussion of concepts and theories is only very loosely related to the discussion of the empirical findings.

Response- We very much appreciate reviewer’s suggestion. We have added theory of governance in section 2.1. From ‘State’ and ‘Market’ to ‘Governance’ and have linked it to the higher education governing system in section 2.2. We have looked at the urban role of universities in the new knowledge economy in section 2.3 and then discussed how university internal governance responds to such external environment and demands in section 2.4. Based on the arguments in these sections, we have developed our conceptual framework in section 2.5 for understanding/exploring governance at three levels of superstructure, middle-structure and understructure.

Methodology and case-study

The article, derived from a PhD thesis, has two weaknesses in my opinion. First, it lacks of clear description of (a) the university and its campuses; and (b) the actual processes of decision making on the issue of their co-location. As a result it is difficult for the reader to get a clear picture. For example, a simple chronology of decisions by the various actors would have helped. Secondly, the article is based on interviews with key actors. Although this is a decent number, no indication of how many interviews were conducted with which types of actors (and the basis for selecting interviews). Also almost no narrative - direct quotations or summaries of views expressed - is given. Instead there is a series of general comments / conclusions.

Response- We very much appreciate reviewer’s suggestion and we have updated our methodology section, which will be well reflected by the revised version of manuscript. We have inserted a map of campuses and a table of decisive milestones in the history of the Norwegian University of Science and Technology (NTNU)’s co-location process. We have added the position of our interviewees and the year(s) that they were involved in the process. We have also added some quotations to support our arguments.

Conclusion

The overall conclusion is unclear, apart from a restatement of the research question (the respective influences of different governance levels). Little attempt is made to indicate what contribution has been made to new knowledge, either in a theoretical sense or in terms of public policy making.

To some degree these weaknesses arise from the, at times, poor English in which the article is written. It does not matter that odd words or missing or that the grammar is incorrect; this could easily be remedied. More serious is that because of the way in which some ideas are expressed the points being made and the argument developed is sometimes obscure and difficult to follow.

I have reluctantly (because it is an interesting topic and the author has clearly worked hard) come to the conclusion that it is not suitable for publication in anything like its current form.

Response- Thank you so much for your comment. We used MDPI English editing service to check the text for correct use of grammar and common technical terms, and to edit it to a level suitable for reporting research in a scholarly journal. In addition, we have taken the editor’s suggestion on re-organization of the paper and we believe there is a better cohesion between our theory, analysis and conclusion.

Round 2

Reviewer 1 Report

The authors have made a strong revision based on the two reviewers comments. I find that the section on governance could be edited for fluency- (on page 3) - these arguments could be nuanced and placed into the context of a dynamic literature (i.e so and so argues, rather than statements around the development of the governance concept presented as uncontested). With that change, this paper should be published. 

Author Response

Dear Reviewer,

We wish to express our appreciation for your insightful comments, which have helped us significantly to improve our manuscript. According to the suggestions, we have thoroughly revised our manuscript and its final version is enclosed. Point-by-point responses to the comments are listed below.

Reviewer #1:

The authors have made a strong revision based on the two reviewers comments. I find that the section on governance could be edited for fluency- (on page 3) - these arguments could be nuanced and placed into the context of a dynamic literature (i.e so and so argues, rather than statements around the development of the governance concept presented as uncontested). With that change, this paper should be published.

Response- We strongly appreciate your comment on this point. We have taken it in full consideration and the text in page 3 is re-written. The article with track changes is attached. Please see the attachment.

Reviewer 2 Report

This revised article is much improved. The changes / additions the author(s) have made address nearly all the concerns I expressed in my original review of the paper. The local context in Trondheim and the evolution of the university are now properly explained. The research methodology is also described in greater detail, and the inclusion of direct quotations from interviewees is a great improvement. 

I still believe that the research base is too slight to justify some of the more general conclusions about ‘governance’. But this is a common fault with qualitative research unless a large number of interviews are conducted. As it reads now the article is a fascinating case-study of the interplay between institutional, local / regional and national layers of governance. However I still feel its particular characteristics may be more interesting than some of the generalisations suggested by the author(s). In other words is what has happened in Trondheim a special case, perhaps suggestive of wider trends, or does it ‘prove the rule’.

There are still a few oddities in the English, eg ‘who’ rather than ‘which’ in relation to an inanimate object. But they are largely trivial.

In conclusion I believe that, with these changes, the article is worthy of publication in ‘Social Sciences’.

Author Response

Dear Reviewer,

We wish to express our appreciation for your insightful comments, which have helped us significantly to improve our manuscript. According to the suggestions, we have thoroughly revised our manuscript and its final version is enclosed. Point-by-point responses to the comments are listed below.

Reviewer #2:

This revised article is much improved. The changes / additions the author(s) have made address nearly all the concerns I expressed in my original review of the paper. The local context in Trondheim and the evolution of the university are now properly explained. The research methodology is also described in greater detail, and the inclusion of direct quotations from interviewees is a great improvement.

I still believe that the research base is too slight to justify some of the more general conclusions about ‘governance’. But this is a common fault with qualitative research unless a large number of interviews are conducted. As it reads now the article is a fascinating case-study of the interplay between institutional, local / regional and national layers of governance. However I still feel its particular characteristics may be more interesting than some of the generalizations suggested by the author(s). In other words is what has happened in Trondheim a special case, perhaps suggestive of wider trends, or does it ‘prove the rule’.

Response- We agree with the reviewer and we did some changes in reaction to the comment above that can be found in the conclusion section in the attached file ‘Article with trach changes’. In addition, we added this paragraph:

“This paper does not suggest ‘the government’s control functions or hierarchical coordination’ as a clear-cut recipe for dealing with university governance challenges. On the other hand, we have tried to elaborate on the theoretical and empirically founded insights of the governance approach and have provided a recommendation for dealing with the potential deficiencies that can undermine the universities’ societal contribution. We are aware that our ideas require further elaboration that they will not always be practical in other cases, and there may be another solution that we have not foreseen.”

There are still a few oddities in the English, eg ‘who’ rather than ‘which’ in relation to an inanimate object. But they are largely trivial. In conclusion I believe that, with these changes, the article is worthy of publication in ‘Social Sciences’.

Response- Thanks for your positive conclusion. We have taken reviewer’s comment in full consideration and have edited the English errors, so it now reads:

“In the ‘top-down’ type of system, the subordinates merely respond to the superiors’ suggested policy initiatives, who are enforced by their power and authority.”

“According to the journalist of the local newspaper Adresseavisen, who has covered most of the campus development issues.”

“Correspondingly, the university development had extensively mattered to Trondheim Municipality, who initiated a formal city-university collaboration.”
